# Conserved neural circuit structure across *Drosophila* larval development revealed by comparative connectomics

**Stephan Gerhard[1†], Ingrid Andrade[1‡], Richard D Fetter[1§], Albert Cardona[1,2]\*, Casey M Schneider-Mizell[1]\***

[1]Janelia Research Campus, Howard Hughes Medical Institute, Ashburn, United States; [2]Department of Physiology, Development and Neuroscience, University of Cambridge, Cambridge, United Kingdom

**\*For correspondence:**
cardonaa@janelia.hhmi.org (AC);
schneidermizellc@janelia.hhmi.org
(CMS-M)

**Present address:** [†]Friedrich Miescher Institute, Basel, Switzerland; [‡]Department of Molecular, Cell and Developmental Biology, University of California Los Angeles, Los Angeles, United States; [§]Department of Biochemistry and Biophysics, University of California San Fransisco, San Fransisco, United States

**Competing interests:** The authors declare that no competing interests exist.

**Abstract** During postembryonic development, the nervous system must adapt to a growing body. How changes in neuronal structure and connectivity contribute to the maintenance of appropriate circuit function remains unclear. Previously , we measured the cellular neuroanatomy underlying synaptic connectivity in *Drosophila* (*Schneider-Mizell et al., 2016*). Here, we examined how neuronal morphology and connectivity change between first instar and third instar larval stages using serial section electron microscopy. We reconstructed nociceptive circuits in a larva of each stage and found consistent topographically arranged connectivity between identified neurons. Five-fold increases in each size, number of terminal dendritic branches, and total number of synaptic inputs were accompanied by cell type-specific connectivity changes that preserved the fraction of total synaptic input associated with each pre-synaptic partner. We propose that precise patterns of structural growth act to conserve the computational function of a circuit, for example determining the location of a dangerous stimulus.
DOI: https://doi.org/10.7554/eLife.29089.001

## Introduction

As an animal undergoes postembryonic development, its nervous system must continually adapt to a changing body. While developing neural circuits can produce new behaviors, such as the addition of new swimming strategies in zebrafish (*Björnfors and El Manira, 2016*), in many cases circuit function is conserved as an animal grows. For example, as a *Drosophila* larva grows from a first instar just out of the egg to a third instar ready to pupariate, its body wall surface area grows by two orders of magnitude (*Keshishian et al., 1993*). To accommodate this growth, mechanosensory neurons grow their dendrites to maintain receptive fields (*Grueber et al., 2002*), while larval motor neurons add more synapses at the neuromuscular junction and change firing properties to maintain functional responses in much larger muscles (*Keshishian et al., 1993*; *Guan et al., 1996*; *Davis and Goodman, 1998*; *Rasse et al., 2005*). Similar functional maintenance has been observed in central circuits as well, from the frequency selectivity of cricket mechanosensory interneurons (*Murphey and Chiba, 1990*) to motor rhythms in crustacean stomatogastric ganglion (STG) (*Bucher et al., 2005*).

A neuron's function arises from the combination of its morphology, synaptic connectivity, and ion channel properties. If morphology and membrane properties co-vary in precise ways, a neuron's integration properties can be consistent across homologous cells, even between species with very different brain sizes (*Cuntz et al., 2013*). Homeostatic regulation of functional and structural properties has been proposed as a key principle in neuronal development, allowing consistent output in the presence of both growth and an uncertain or ever-changing environment (*Kämper and*

*Murphey, 1994*; *Bucher et al., 2005*; *Marder and Goaillard, 2006*; *Tripodi et al., 2008*; *Giachello and Baines, 2017*).

It remains unclear how neuronal circuits adapt during development by changing their anatomical structure — varying size, adding branches, or producing new synaptic connections — as opposed to adaptation in intrinsic functional properties like ion channel expression and distribution. Studies of circuit variability offer hints, since variability reflects differences in the outcomes of neurons following the same developmental rules. For rhythmic pattern generator circuits, similar temporal dynamics can be produced in many different ways. Simulations of STG have found that numerous different combinations of intrinsic functional parameters and synaptic weights are able to produce extremely similar dynamics (*Grashow et al., 2010*; *O'Leary et al., 2014*; *Prinz et al., 2004*). Correspondingly, the morphological structure of neurons (*Otopalik et al., 2017*) and their functional connection strengths (*Goaillard et al., 2009*) have been observed to have high inter-animal variability while still generating similar motor patterns. Observations of inter-animal variability in leach heartbeat networks (*Norris et al., 2011*; *Roffman et al., 2012*) suggest that this may be a general principle for rhythm generating circuits.

However, synaptic-resolution electron microscopy (EM) reconstructions from *Drosophila* sensory systems have found relatively low intra-animal variation in number of synaptic contacts between columnarly repeated neurons in the adult visual system (*Takemura et al., 2015*) or bilaterally repeated neurons in the mechanosensory (*Ohyama et al., 2015*), visual (*Larderet et al., 2017*), and olfactory (*Berck et al., 2016*) systems. Comparisons between individuals at this scale have been limited due to incomplete image volumes (*Ohyama et al., 2015*) or high error rates with early reconstruction methods (*Takemura et al., 2013*).

Here, we used detailed circuit reconstruction from EM to study the circuitry of identified neurons across postembryonic development in two *Drosophila* larvae. Despite considerable growth in body size between hatching and pupariation, almost no new functional neurons are added to the larval nervous system (*Truman and Bate, 1988*) and behavior remains largely unchanged (*Almeida-Carvalho et al., 2017*). Nonetheless, electrophysiological and light microscopy analysis has shown that central neurons become larger (*Zwart et al., 2013*) and have more synapses both in total (*Zwart et al., 2013*) and in specific connections (*Couton et al., 2015*).

## Results

### mdIV axon terminals increase in size and number of synapses

We focused on nociception, a somatosensory modality crucial for larvae to avoid wide-ranging sources of damage, such as parasitoid wasp attack (*Hwang et al., 2007*; *Robertson et al., 2013*) or intense light (*Xiang et al., 2010*). Nociceptive stimuli are detected by the three multidendritic class IV sensory neurons (mdIVs) (*Hwang et al., 2007*) in each hemisegment, with dendrites that tile the body wall (*Figure 1A*) (*Grueber et al., 2002*). We began by investigating the structure of the mdIV axon terminals at first and third instar stages. The mdIV terminals in abdominal segment A1 of an early first instar larva EM volume (L1v) were previously reconstructed (*Ohyama et al., 2015*). We generated a new serial section transmission EM volume of a late third instar larva (L3v) spanning several abdominal segments of the ventral nerve cord (VNC) (*Figure 1—figure supplement 1A,B*). In the L3v, we manually reconstructed the six mdIV terminals (three per hemisegment) in abdominal segment A3 (*Figure 1B,C* and *Figure 1—figure supplement 1C*) using the web-based tool CAT-MAID (*Saalfeld et al., 2009*; *Schneider-Mizell et al., 2016*). Segment A3 was chosen due to its centrality in the L3v and lack of image artifacts or missing sections. In all cases, we reconstructed neurons as skeletons, expressing the 3D topology of neuronal arbors, but not their diameter or volume. The identity of each mdIV terminal was determined based on stereotyped morphological features such as antero-posterior projections, midline crossing, and nerve bundle (*Figure 1B,C* and *Figure 1—figure supplement 2*) (*Merritt and Whitington, 1995*; *Grueber et al., 2007*; *Ohyama et al., 2015*).

The morphology of mdIV axon terminals remained similar across larval stages, with growth in overall size but no change in branching patterns (*Figure 1B–D* and *Figure 1—figure supplement 2*).

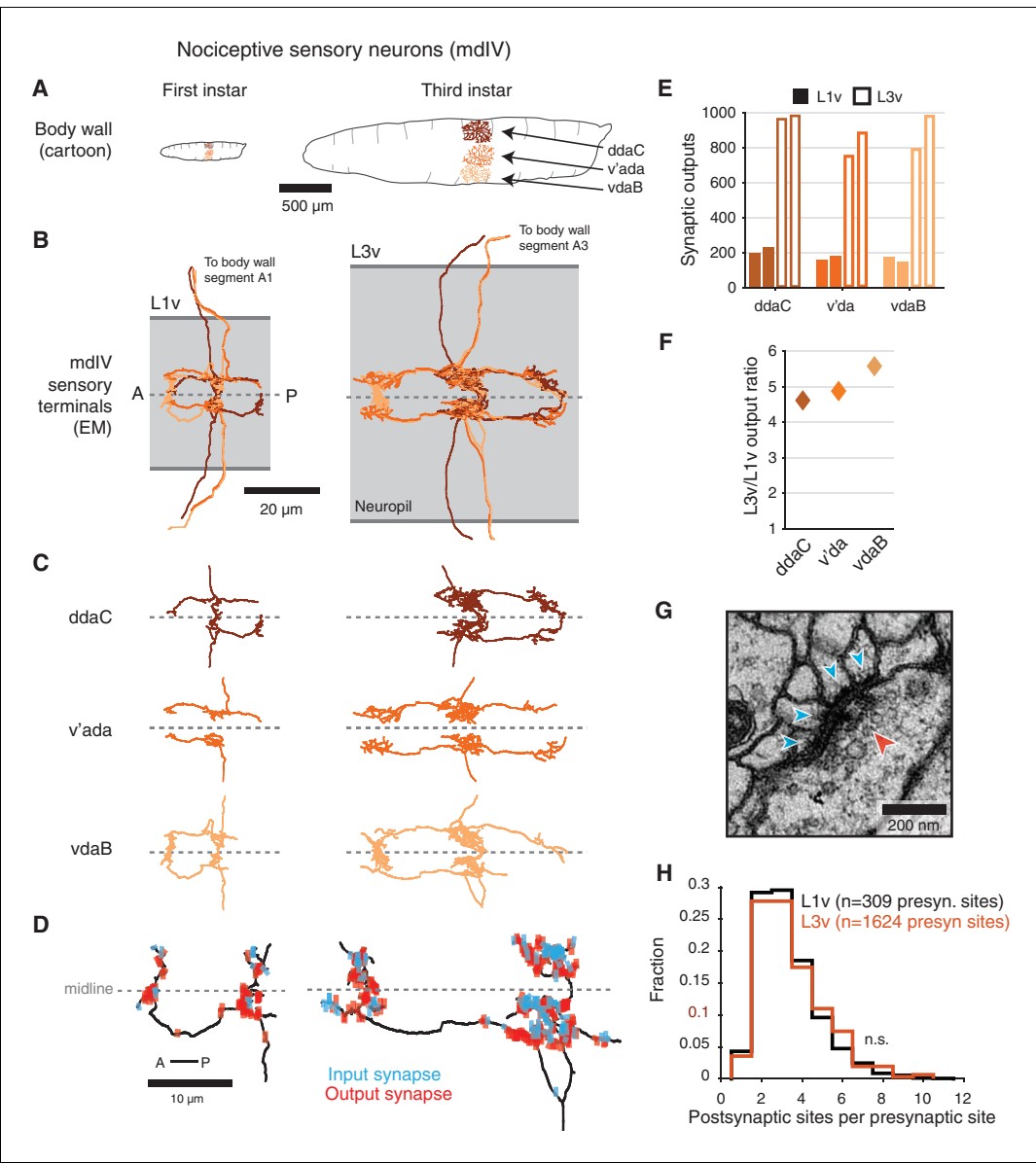

**Figure 1.** Structure of mdIV terminals through postembryonic development. (**A**) Cartoon comparison of the dendritic fields of the three nociceptive mdIV sensory neurons from a single hemisegment at first and third instar stages; sagittal view; anterior to left. (**B**) Dorsal view of EM reconstructions of all mdIV terminals from a single abdominal segment in the L1v (left; segment A1, first instar larva) and L3v (right; segment A3, third instar larva) data. Colors are as in **A**. The vertical extent of the gray box indicates the width of the neuropil; anterior to left; dashed line indicates midline. (**C**) Morphology of the terminals of each mdIV subtype, presented as in **B**. Unbranched primary projections from the nerve are cropped. (**D**) Dorsal view of a single vdaB terminal from the L1v and L3v, shown with synapses (outputs, red; inputs, cyan). Dashed line indicates midline. (**E**) Number of synaptic outputs on each mdIV terminal. L1v (solid bars), L3v (empty bars); left/right bar corresponds to left/right neuron. (**F**) Fold-change in synaptic outputs in the L1v and L3v. For each mdIV subtype, left and right neurons were averaged. (**G**) A standard polyadic synapse. In this example, taken from the L3v, the single pre-synaptic site (red arrowhead) has four post-synaptic contacts (cyan arrowheads). (**H**) Normalized histogram of number of post-synaptic contacts per pre-synaptic site on mdIV terminals (No significant difference; p=0.5641, two-sided Kolmogorov-Smirnov test). n.s. not significant; *p<0.05. **p<0.01. ***p<0.001.

DOI: https://doi.org/10.7554/eLife.29089.002

The following figure supplements are available for figure 1:

**Figure supplement 1.** A new EM image volume from a third instar larva ventral nerve cord.

DOI: https://doi.org/10.7554/eLife.29089.003

*Figure 1 continued on next page*

*Figure 1 continued*

**Figure supplement 2.** Reconstructions of mdIV terminals.

DOI: https://doi.org/10.7554/eLife.29089.004

However, the number of synaptic outputs increased by a factor of 4.7, from a mean of 185 synapses per terminal to 872 synapses per terminal (*Figure 1E,F*). Insect synapses are polyadic, with multiple post-synaptic targets per pre-synaptic site (*Figure 1G*), thus this increase could arise from either changes in the number of distinct pre-synaptic sites or the number of targets per pre-synaptic site. We found no significant difference between the distribution of number of post-synaptic targets for mdIV pre-synaptic sites in the L1v compared to the L3v (*Figure 1H*), suggesting the structure of individual polyadic synapses remains unchanged.

## Nociceptive interneurons increase in total dendritic cable length and synaptic inputs

The pattern of sensory input onto second-order interneurons is a key component of early sensory processing. To comprehensively identify all second-order mdIV neurons in the L1v, we used all pre- or post-synaptic contacts with mdIV terminals to seed further reconstructions (*Figure 2A*). We found

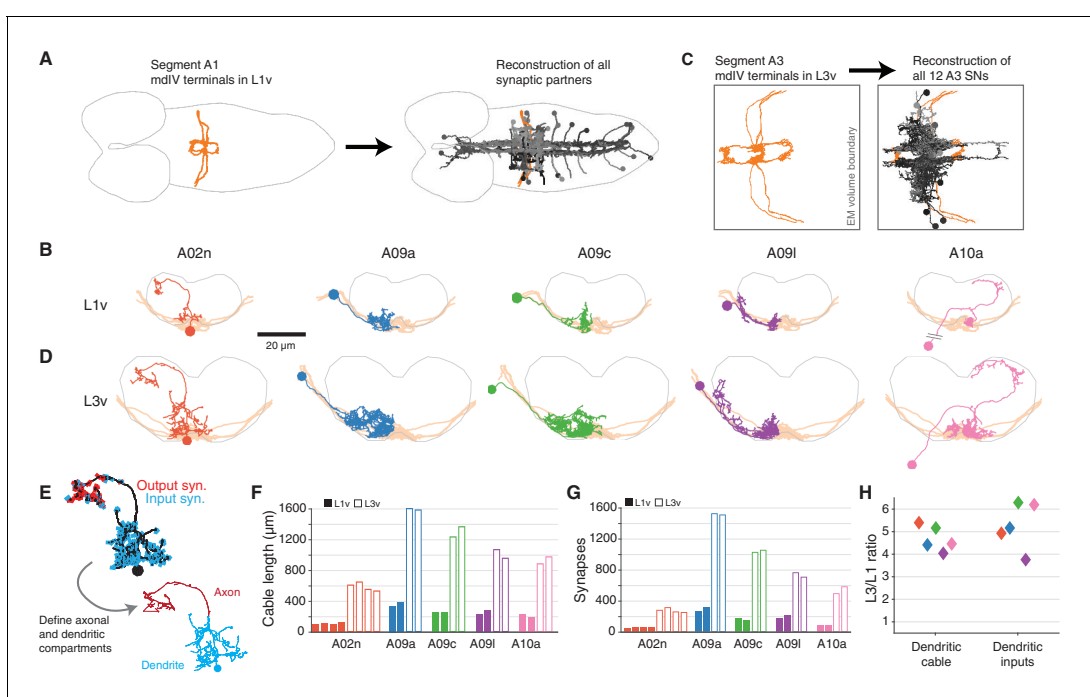

**Figure 2.** Morphology and properties of second-order nociceptive LNs. (**A**), Starting from the synapses of the mdIVs in segment A1 of the L1v, we reconstructed all synaptic partners (grays). See *Figure 2—figure supplement 1* for details of each cell type. Dorsal view, gray outline indicates CNS boundary; anterior is to left. (**B**) Examples of the anatomy of all five classes of LNs from the L1v. Posterior view; gray outline indicates neuropile boundary, orange shows mdIV position. (**C**) Based on the mdIV reconstructions in the L3v (orange), we reconstructed the same populations of all mdIV LNs in segment A3 (grays; 12 LN cells in total). (**D**) Examples of the anatomy of all five classes of LNs from the L3v, shown as in B. (**E**) All neurons were split into axonal and dendritic compartments based on well-separated synaptic input and output domains. The example shown is the A02n from **D**. (**F**), Total dendritic cable length for all LNs. (**G**) Number of synaptic inputs onto LN dendrites. (**H**) Fold-change in dendritic cable length and dendritic synaptic inputs between the L1v and L3v LNs.

DOI: https://doi.org/10.7554/eLife.29089.005

The following figure supplements are available for figure 2:

**Figure supplement 1.** The complete second-order mdIV network from the L1v.

DOI: https://doi.org/10.7554/eLife.29089.006

**Figure supplement 2.** Additional LN properties.

DOI: https://doi.org/10.7554/eLife.29089.007

that there are 13 distinct cell types stereotypically connected to mdIV terminals (*Figure 2—figure supplement 1A–C*). Five types were local neurons (LNs), with dendrites covering 1–2 segments (*Figure 2B*); three were regional, with dendrites covering 3–5 segments; four were ascending neurons projecting across the entire VNC; and one was a descending neuron with an axon that projected along the whole VNC (*Figure 2—figure supplement 1A*). One cell type (A02n) was comprised of two indistinguishable cells per hemisegment, unusual for the larva, making a total of twelve LNs cells per segment. Note that two LNs, A09a and A09c, have been the focus of previous work under the names 'Basin-2' and 'Basin-4' (*Ohyama et al., 2015*; *Jovanic et al., 2016*). Second-order nociceptive interneurons formed a sparse network (*Figure 2—figure supplement 1B,C*), without the densely connected local interneurons found in other early processing of other *Drosophila* sensory modalities like olfaction (*Berck et al., 2016*; *Liu and Wilson, 2013*) or mechanosensation (*Jovanic et al., 2016*; *Tuthill and Wilson, 2016*).

To measure how second-order nociceptive interneurons change across larval growth, we reconstructed all twelve third instar LNs in the L3v (*Figure 2C*). Each LN was morphologically identifiable, despite increases in size, arbor complexity, and synaptic count (*Figure 2B,D*). For every LN, the spatial segregation of synaptic input and synaptic output made it possible to split neuronal arbors into a separate dendritic domain and axonal domain (*Figure 2E*). Dendritic cable length, defined as the sum total length of all dendritic neurites, increased by an average factor of $4.69 \pm 0.28$, consistent with the increase measured from light microscopy in larval motor neurons (*Zwart et al., 2013*) (*Figure 2F,H*). The number of synaptic inputs onto LN dendrites increased similarly, by an average factor of $5.28 \pm 0.52$ (*Figure 2G,H*). Only three LN types had axons fully contained in the L3v (A02n, A08l, and A10a), but our data suggest that axons and dendrites differed in their overall morphological growth (*Figure 2—figure supplement 2A–E*). In particular, axonal cable length increased by an average factor of only $2.15 \pm 0.33$, significantly less than the scale-up of dendritic cable (*Figure 2—figure supplement 2A,E*), and close to the overall 1.7–1.8 times scale-up of neuropile width (L1v: 43 $\mu$m; L3v, 72 $\mu$) and segment length (L1v, 15 $\mu$m; L3v, 27 $\mu$m). Only those LN types that exhibited dendritic outputs in the L1v also did so in the L3v (*Figure 2—figure supplement 2D*). For each LN the broad pattern of segregation between synaptic inputs and outputs, which indicates the degree of local dendritic output and axonal pre-synaptic input, was preserved over postembryonic development (*Figure 2—figure supplement 2F*).

## Nociceptive interneurons maintain a topographically-arranged distribution of mdIV synaptic inputs across larval development

Since both mdIV terminals and LN dendrites grow more synapses, we next measured how the synaptic connectivity from mdIVs onto LNs changed across larval development. Every LN in the L3v received synaptic input from mdIVs (*Figure 1—figure supplement 1D–F*). On average, the total count of synaptic input from mdIVs differed by a factor of $5.77 \pm 1.11$ from the L1v (*Figure 3A,C*). However, the normalized synaptic input, defined here as the number of synapses in a connection divided by the total number of dendritic input synapses on the post-synaptic neuron, remained strikingly stable, changing by an average factor of $1.09 \pm 0.20$ (*Figure 3B,C*).

LNs do not receive synaptic input equally from all mdIV subtypes. The normalized synaptic input into each LN from each mdIV axon was highly structured in both L1v and L3v data (*Figure 3D*). For each mdIV type and LN type, the average normalized synaptic input was significantly correlated between L1v and L3v (*Figure 3E*). Moreover, the variability between left and right cells of the same type was significantly lower in the L3v than the L1v (*Figure 3F*). These observations suggest that there is, effectively, a target value for the normalized synaptic input for each connection and this value is achieved more precisely as the nervous system develops postembryonically.

We speculated that the ability to respond according to location of stimuli on the body wall is likely to be an important conserved function of mdIV circuitry. Each segment of the body wall is spanned by six mdIVs whose dendritic fields divide the left and right sides into dorsal, lateral, and ventral thirds (*Figure 3G*) (*Grueber et al., 2002*). For each LN, we approximated the mean orientation of its input as the average of unit vectors oriented toward the center of each mdIV dendritic field, weighted by its associated synaptic count (*Figure 3H*) (see Materials and methods). We found that LN orientations span the body wall, and the orientation of LNs are conserved across development. Further, LN inputs are arranged so that a nociceptive stimulus smaller than a single mdIV's dendrite, for example a wasp ovipositor (*Robertson et al., 2013*), is likely to drive different

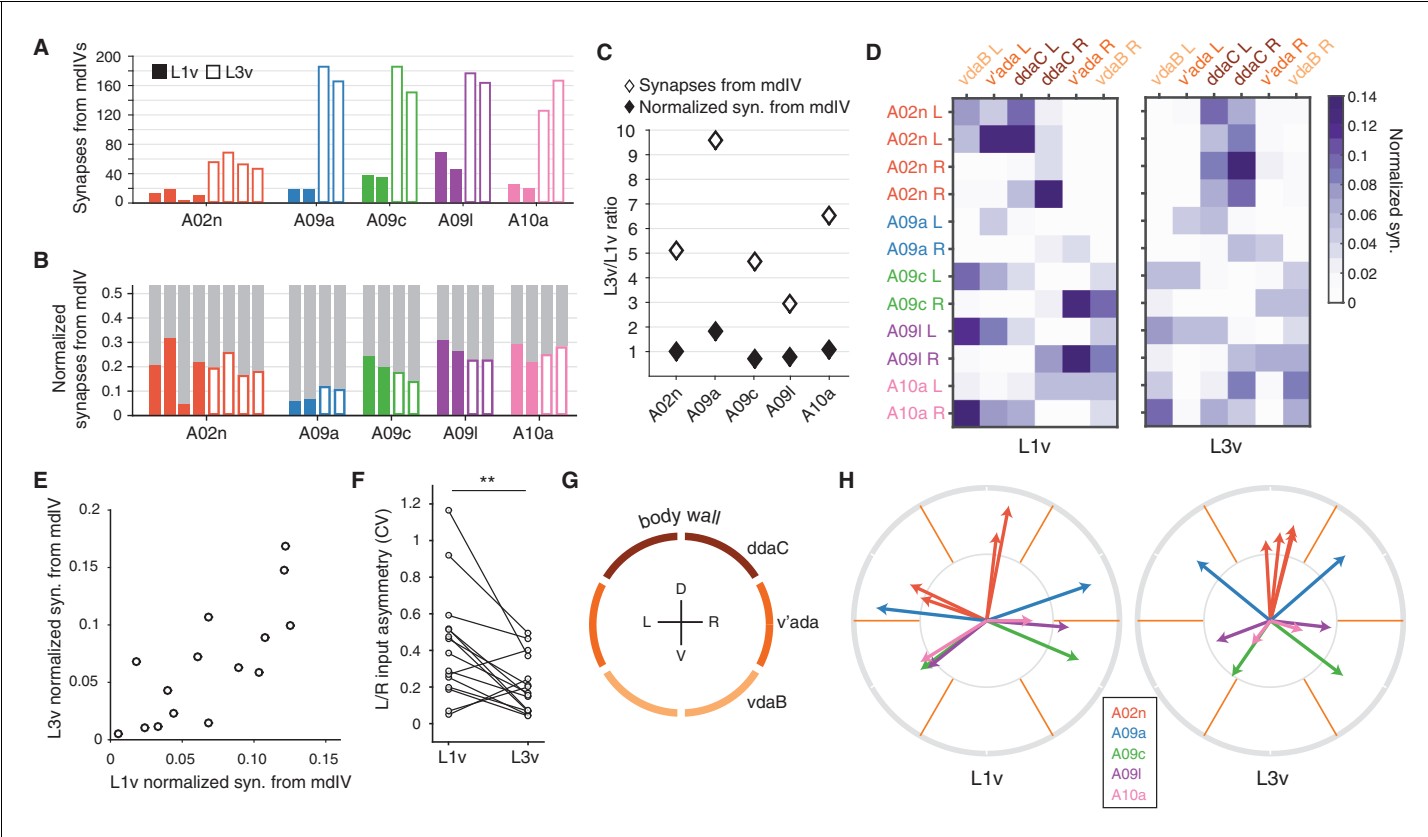

**Figure 3.** Connectivity of second-order nociceptive LNs is topographically arranged and consistent across larval development. (A) Number of synaptic inputs onto LNs from mdIV terminals in the same segment. (B) Normalized dendritic synaptic input from mdIV terminals for each LN. (C) Fold-change in number of synapses and normalized synaptic inputs from mdIVs for each LN type. (D) Heatmap of normalized dendritic input from each mdIV terminal onto each LN for L1v (left) and L3v (right). Note that mdIV terminals are ordered clockwise from ventral left. (E) Normalized dendritic input from mdIVs onto LNs is strongly correlated across animals and developmental time points. Each data point corresponds to average normalized dendritic input from an mdIV type onto an LN type. (Pearson's r = 0.77, p<0.001 to be different from zero). (F) Asymmetry between normalized mdIV synaptic input into left and right LNs, measured as coefficient of variation. Asymmetry in the L3v is significantly lower (p=0.006, paired two sided t-test). (G) Cartoon of the larval body wall viewed from posterior. The dendritic receptive field of each mdIV covers approximately 1/6 of the circumference of the animal. (H) Mean body wall orientation of mdIV input into each LN in the L1v (left) and L3v (right), computed as the average of unit vectors pointing at the center of each mdIV dendrite receptive field, weighted by number of synaptic inputs from that neuron. Arrow color corresponds to LN type. n.s. not significant; *p<0.05. **p<0.01. ***p<0.001.

DOI: https://doi.org/10.7554/eLife.29089.008

The following figure supplement is available for figure 3:

**Figure supplement 1.** Topographically structured feed-forward connectivity between mdIV-related LNs.

DOI: https://doi.org/10.7554/eLife.29089.009

populations of LNs based on its exact location, with the smallest difference being between left and right ventral regions (*Figure 3D*). Interestingly, only LNs with similar input orientations synaptically connect to one another (*Figure 3—figure supplement 1*), suggesting that convergent feed-forward motifs are specifically present within sets of neurons likely to be driven at the same time. Conservation of synaptic input through larval development thus preserves the topographical structure of the nociceptive circuit, both in sensory input and interactions between interneurons.

## The likelihood of synaptic contact between nearby neurons is stereotyped, cell type-specific, and conserved across larval development

To better understand how synaptic and morphological changes work together to maintain specific patterns of normalized synaptic input, we analyzed the relationship between the spatial location of

neuronal arbors and their connectivity. A post-synaptic neuron can only connect to pre-synaptic sites that are nearby in space, or 'potential synapses'. Numerically strong connections could arise either due to a low probability of connecting to many nearby potential synapses, or to a high probability of connecting to fewer potential synapses. To distinguish these scenarios, we measured 'filling fraction' (*Stepanyants et al., 2002*), defined as the fraction of potential synapses that are actually connected (*Figure 4A,B*) (see Methods). In both the L1v and L3v, filling fraction ranged from 0.01 to 0.47, indicating that some connections from mdIV types to LNs were realized much more often than others. Filling fraction correlated strongly with the overall count of synapses in a connection, suggesting that numerically strong connections are produced through high connection probability, not only increased potential synapse counts (*Figure 4C,D*). Moreover, filling fraction was significantly correlated between the L1v and L3v (*Figure 4E*), suggesting that the local propensity to form stable synapses with a nearby cell type is preserved across development.

Post-synaptic connections are not evenly distributed throughout a neuron's dendrite. Most synaptic input onto a neuron is located on 'twigs', spine-like microtubule-free terminal branches hosting a small number of synapses, in contrast to the microtubule-containing 'backbone' that spans the soma and all of the main branches of a neuron (*Leiss et al., 2009*; *Schneider-Mizell et al., 2016*) (*Figure 5A,B*). In order to host an increased number of synaptic inputs, a neuron's twigs would need

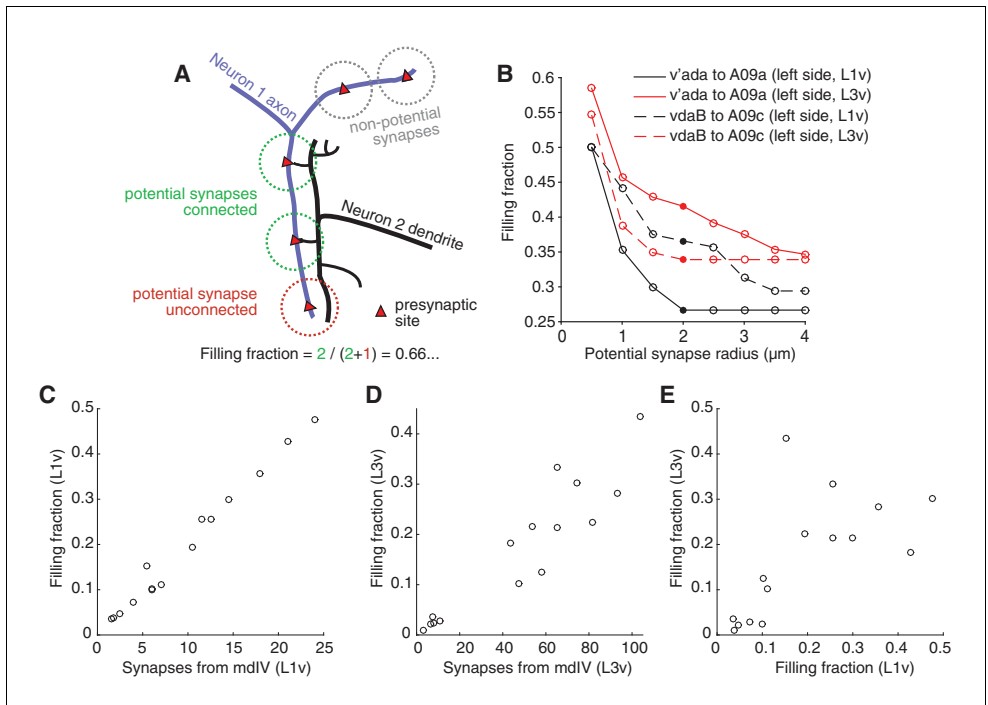

**Figure 4.** Numerically strong connections are associated with stereotypically high filling fraction. (**A**) Description of 'filling fraction' for a connection from Neuron 1 (purple) to Neuron 2 (black). Neurons can only be connected where they are adjacent to one another in space. A pre-synaptic site on Neuron 1 is a potential synapse from Neuron 1 to Neuron 2 if any part of Neuron 2 passes within a given radius (dashed circles). Filling fraction is defined as the number of potential synapses (red and green dashed circles) that are actually connected (green dashed circles only). (**B**) Dependence of filling fraction on the potential synapse radius for four example connections. For subsequent figures, we chose 2 $\mu$m (filled circles) as a compromise between the typical size of a terminal branch and a shoulder in the filling-fraction versus radius curve. (**C–D**) Mean filling fraction vs. mean number of synapses in the L1v (**C**) and L3v (**D**). Each data point represents the average value for connections from mdIV types onto LN types. The high correlation in both (L1v, Pearson $r$ = 0.99, p<0.001 different from zero; L3v, Pearson $r$ = 0.93, p<0.001) suggests that increased connection probability, not merely access to differing numbers of pre-synaptic synapses, helps set cell type-specific differences in synaptic counts. (**E**) Filling fraction of mdIV type to LN type connections in the L1v and L3v are significantly correlated with one another (Pearson $r$ = 0.64, p=0.009).

DOI: https://doi.org/10.7554/eLife.29089.010

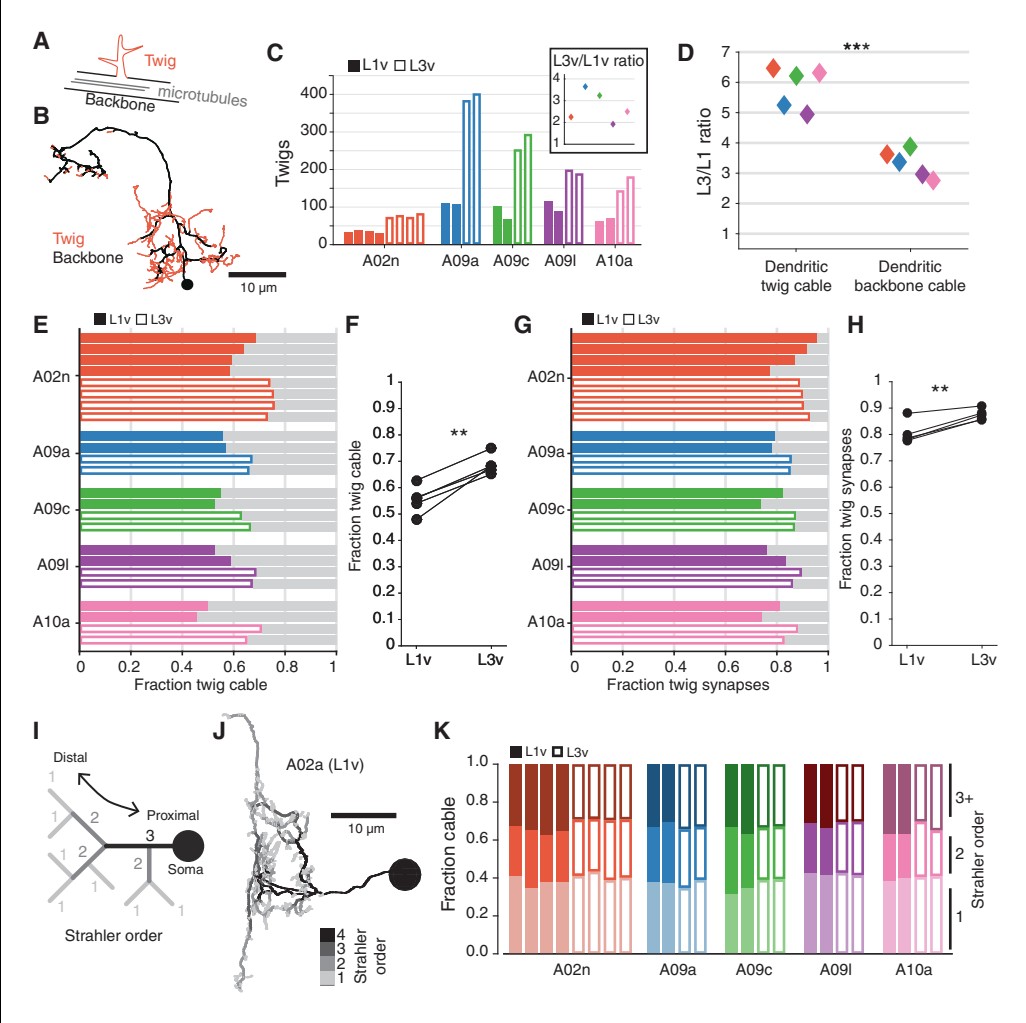

**Figure 5.** The structure of terminal dendritic branches across postembryonic development. (**A**) Definition of microtubule-containing 'backbone' (black) and microtubule-free, spine-like 'twigs' (red). (**B**) Example A02n cell (from the L3v) where all twigs are labeled, posterior view. (**C**) Number of twigs in each LN in the L1v and L3v. Inset: Fold-change in number of twigs between the L1v and L3v. (**D**) Fold-change in length of cable comprised of twigs or backbone in the L1v and L3v. Twigs increase more than backbone (two sided t-test). (**E**) Fraction of dendritic cable comprised of twigs for all LNs. (**F**) The average fraction of dendritic cable comprised of twigs per cell type was larger in L3v than L1v (two sided, paired t-test). (**G**) Input synapses that contact twigs as a fraction of all input synapses for all LNs. (**H**) The fraction of input synapses that are onto twigs increased significantly (two-sided, p=0.003, paired t-test). (**I**) Cartoon definition of Strahler order. Terminal tips are defined to have Strahler order 1. Where two branches with the same Strahler order converge, the value increments by one. The most core, proximal neurites thus have the highest Strahler order. (**J**) An example A09a cell from the L1v with branches labeled by Strahler order (Dorsal view). (**K**) Fraction of dendritic cable for each LN cell by Strahler order. The relative amount of cable with low Strahler order (i.e. distal) is approximately conserved between the L1v and L3v neurons. n.s. not significant; *p<0.05. **p<0.01. ***p<0.001.

DOI: https://doi.org/10.7554/eLife.29089.011

The following figure supplement is available for figure 5:

**Figure supplement 1.** Twig and backbone morphology for all LNs.

DOI: https://doi.org/10.7554/eLife.29089.012

to change, growing more twigs or hosting more synapses per twig. To measure this, we manually identified all twigs in the twelve LNs (*Figure 5—figure supplement 1*). We found that the number of twigs increased by an average factor of $2.70 \pm 0.36$ (*Figure 5C*). The total length of dendritic cable that twigs span increased by a factor of $5.85 \pm 0.31$, significantly more than dendritic backbone

$(3.31 \pm 0.20)$ (*Figure 5D*). Both the fraction of dendritic cable comprised of twigs (*Figure 5E,F*) and the fraction of dendritic input synapses onto twigs (*Figure 5G,H*) increased significantly, suggesting that twigs become even more central to dendritic input.

## An increased number of small twigs host a larger fraction of synaptic input

Measuring twigs requires painstaking visual inspection of EM imagery, so we also looked at a purely topological measure of neuronal arbor structure, Strahler order (*Binzegger et al., 2005*), that matches intuitive definitions of proximal and distal branches (*Figure 5I,J*). We found that the fraction of dendritic cable that is last or next-to-last (Strahler order 1 or 2) order is similar not only across development, but also across cell types (*Figure 5K*). For this observation to be consistent with the relative increase in dendritic twigs for cable, the properties of individual twigs must change so that twigs in the L3v have branches with higher Strahler order than in the L1v. This suggests that in the larva, neurons grow their dendrites by both increasing the number of twigs, while also modestly increasing the length of the backbone neurites from which they sprout.

## Twigs remain short and continue to host few synaptic inputs

To get better insight into how twigs changed between the first and third instar, we measured the properties of individual twigs on LNs. Typical dendritic twigs in both the L1v and L3v are small. They were short in both total length and maximum depth from twig root, had few branch points, and few post-synaptic sites (*Figure 6A*). However, twigs in the L3v were slightly longer than their L1v counterparts and had significantly more branch points (*Figure 6A*). The median distance between adjacent twigs along neuronal backbone remains similar (L1v: 0.83 $\mu m$; L3v: 1.03 $\mu m$), suggesting that the density of twigs on branches remains similar even as neurons grow. In a few cases, we also found that there were quantitative differences between the twig properties of different cell types (e.g. A02n twigs were significantly longer and had greater maximum depth than those of other LNs *Figure 6—figure supplement 1*), suggesting that individual cell types can deviate from the typical case.

## The number of distinct twigs involved in a connection increases with the number of synaptic contacts

We next asked how the input from a pre-synaptic sensory neuron is distributed across the twigs on an LN's dendrite for each mdIV→LN connection. Consistent with previous work in the first instar motor system (*Schneider-Mizell et al., 2016*), mdIV→LN connections with many synaptic contacts were distributed across many twigs in both the L1v and L3v — approximately one twig for every 2.5 synapses in a connection in the L3v (*Figure 6B*). Within a single mdIV→LN connection, the vast majority of twigs (L1v: 92.5%; L3v: 81%) hosted only 1 or 2 of the many possible synaptic contacts (*Figure 6C*).

A practical consequence of numerically strong but anatomically distributed synaptic connectivity is that EM reconstruction becomes robust to random errors. The vast majority of manual errors in previous larval reconstructions was the omission of single dendritic twigs (*Schneider-Mizell et al., 2016*). To measure how twig omission rate would affect accuracy in measuring mdIV input into LNs, we simulated the effect of removing random twigs from LN reconstructions. For each mdIV→LN connection, we simulated removing twigs from our anatomical reconstructions with a given omission probability between 0 and 1 (N = 5000 simulations per value), and measured the fraction of synapses that would remain observed in the resulting arbor (*Figure 6D,E*). For concreteness in comparing connections, we found the maximum error rate for which the probability of detecting fewer than 25% of the observed synapses was ≤5% (*Figure 6E*). The anatomical and numerical redundancy of synapses on LN dendrites resulted in connections that would be detectable with at least 25% of the actual number of synapses, even if twigs were missed at the same rate (12%) as observed in previous work with the same reconstruction method (*Figure 6E*). Numerically strong connections in the L3v were particularly robust, and would still be detectable with a 50% false negative rate (*Figure 6E*). For large neurons, a strategy of incomplete sampling could thus quickly identify numerically strong synaptic partners at the cost of precise measurement of synaptic count.

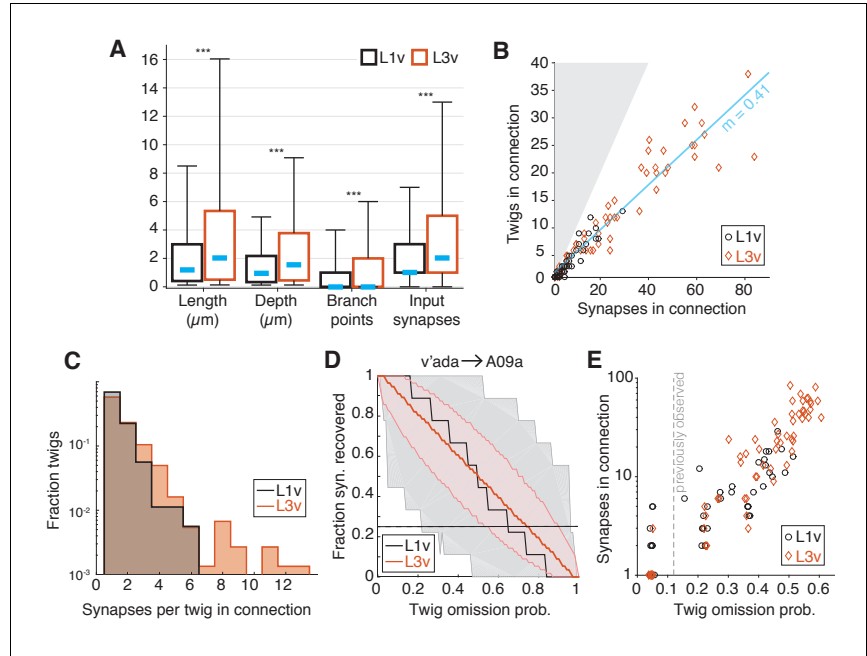

**Figure 6.** Twig properties across postembryonic development. (**A**) Distribution of morphological and synaptic properties of distinct twigs in the L1v and L3v LNs. Total twig length and maximum twig depth are in $\mu$m, branch points and synapses are integer. Boxes are interquartile intervals, blue dashes are median values, and whiskers correspond to 5/95 percentiles. Wilcoxon rank sum test with Bonferroni correction. (**B**) Number of distinct twigs in a connection versus number of synapses in the same connection. The blue line indicates a linear fit to connections with five or more synapses (slope shown). The gray region corresponds to the disallowed situation of more twigs connected than synapses. (**C**) Histogram of number of synapses per twig in each mdIV→LN connection. (**D**) The fraction of synapses in the connection from a v'ada to an A09a in the L1v (9 synapses onto twigs) and L3v (56 synapses onto twigs) recovered after simulated random omission of twigs (N = 5000 instances), as a function of omission probability. Thick lines show median value, shaded region the 5/95 percentile value. The dashed horizontal line indicates the 25% of synapses recovered. (**E**) Maximum error rate permitting the recovery of 25% of synapses with probability >0.05 for each observed mdIV→LN connection (i.e. where the horizontal dashed line crosses into the shaded area in **D**). Each data point is a single mdIV axon's synapses with a single LN. The vertical line indicates the error rate for twigs found previously for manual annotation of motor neurons (*Schneider-Mizell et al., 2016*). n.s. not significant; *p<0.05. **p<0.01. ***p<0.001.

DOI: https://doi.org/10.7554/eLife.29089.013

The following figure supplement is available for figure 6:

**Figure supplement 1.** Individual twig properties, broken down by LN cell type.

DOI: https://doi.org/10.7554/eLife.29089.014

## Discussion

We have shown how in *Drosophila* neuronal arbor morphology changes across postembryonic development while circuit connectivity properties remain largely unchanged. Our findings establish a quantitative foundation for the previous observation that numerically strong connectivity in the L1v predicted the presence of functional connectivity in third instar larvae tested experimentally (*Ohyama et al., 2015*; *Zwart et al., 2016*; *Fushiki et al., 2016*; *Heckscher et al., 2015*; *Jovanic et al., 2016*). In all neurons measured, the basic anatomical elements of connectivity — polyadic synapses and small post-synaptic twigs — remained similar, while neurons grew five-fold in total synaptic input and cable length. For the highly stereotyped, numerically strong mdIV→LN connections, the number of synaptic contacts scaled almost identically to the total number of inputs, suggesting the fraction of total inputs per connection is a developmentally conserved value. Interestingly, although cell types ranged considerably in size at any given time point, the fold-increase in total cable length and synapse count was nearly constant across cell types. We note that the sensory

connections we focused on here are excitatory (*Ohyama et al., 2015*). An interesting avenue of future work would be to examine if inhibitory connections follow similar developmental rules.

## Compensatory changes in synaptic connectivity and the maintenance of circuit function

The tight control of normalized synaptic input is likely to be in the service of circuit function. Our data suggest that, as neurons grow, there is a consistent compensatory growth in synaptic inputs from sensory neurons. This observation suggests that central neurons adapt structurally to compensate for increasing volume with concurrent increases in excitatory synaptic currents by adding synaptic contacts, as seen at the neuromuscular junction (*Rasse et al., 2005*). It is possible that such structural changes are also accompanied by functional changes, for example in neurotransmitter receptor or release properties.

Neuronal computations depend on how dendrites integrate synaptic inputs. In visual system interneurons in the adult fly, dendritic geometry and membrane properties work together so that, near the spike initiation zone, the functional weight of a synaptic input does not depend strongly on its location on the dendrite (*Cuntz et al., 2013*). Similarly, simulations based on adult *Drosophila* olfactory projection neurons reconstructed from EM found that the functional responses were simply proportional to the number of synapses activated, even after shuffling input locations (*Tobin et al., 2017*). Taken together, this suggests linear dendritic integration of excitatory input may be common, at least in early sensory processing. In our data, each mdIV input into LNs typically increased by a common factor, irrespective of specific pre-synaptic cell type. Linear integration would thus imply that the relative functional weight of each mdIV type is preserved across development. The higher scaling of synaptic count in the numerically weakest connections (e.g. mdIV→A09a) could potentially reflect small deviations from linear integration for low numbers of synaptic input.

The same developmental rules that allow neurons to maintain circuit function as the body grows would also be well-suited to handle natural variability, for example from reduced growth due to food restriction (*Mirth and Riddiford, 2007*). Indeed, it is possible that the use of consistent homeostatic rules for cell type-specific connectivity and integration could allow circuits to remain functionally or computationally similar over large evolutionary changes in neuron size. Such homology has been observed in *Calliphora* and *Drosophila* visual system neurons, which were found to differ in scale by a factor of four in each spatial dimension but have retained similar electrotonic structures (*Cuntz et al., 2013*).

Stringent structural stereotypy observed here stands in contrast to rhythm-generating circuits in other invertebrates, in which large variability can be found in morphological and functional properties (*Goaillard et al., 2009*; *Norris et al., 2011*; *Roffman et al., 2012*; *Otopalik et al., 2017*). One possibility is that the computation of certain features from sensory input imposes tighter constraints on circuit structure than the production of periodic activity. The ability to combine detailed measurements of structure with cell type-specific genetic reagents (*Pfeiffer et al., 2008*; *Pfeiffer et al., 2010*) will allow this hypothesis to be tested across different circuits in the fly and to better elucidate the detailed mechanisms underlying their structural development.

## Materials and methods

### Sample preparation and electron microscopy

The L1v is fully described in (*Ohyama et al., 2015*). In brief, the central nervous system from a 6 hr old [iso] *Canton S G1* x [iso] $w^{1118}5905$ female larva were dissected and, after chemical fixation, stained *en bloc* with 1% uranyl acetate, dehydrated, and embedded in Epon resin. Serial 50 nm sections were cut and stained with uranyl acetate and Sato's lead (*Sato, 1968*). Sections were imaged at 3.8×3.8 nm using Leginon (*Suloway et al., 2005*) on an FEI Spirit TEM (Hillsboro). Images were montaged in TrakEM2 (*Saalfeld et al., 2010*; *Cardona et al., 2012*) and aligned using elastic registration (*Saalfeld et al., 2012*).

For the L3v, the central nervous systems from a 96 hr wandering third instar [iso] *Canton S G1* x [iso] $w^{1118}5905$ larva was dissected in PBS and immediately transferred to 125 $\mu$l of 2% glutaraldehyde in 0.1 M Na cacodylate buffer, pH 7.4 in a 0.5 dram glass vial (Electron Microscopy Sciences, cat. no. 72630–05) on ice. 125 $\mu$l of 2% OsO$_4$ in 0.1 M Na-cacodylate buffer, pH 7.4 was then added

and briefly mixed immediately before microwave assisted fixation on ice conducted with a Pelco Bio-Wave PRO microwave oven (Ted Pella, Inc.) at 350W, 375W and 400W pulses for 30 s each, separated by 60 s intervals. Samples were rinsed 3 × 30 s at 350W with 0.1 M Na-cacodylate buffer, separated by 60 s intervals, and post-fixed with 1% OsO$_4$ in 0.1 M Na-cacodylate buffer at 350W, 375W and 400W pulses for 30 s each, separated by 60 s pauses. After rinsing with distilled water 3 × 30 s at 350W with 60 s pauses between pulses, the samples were stained *en bloc* with 7.5% uranyl acetate in water overnight at 4°C. Samples were then rinsed 3 × 5 min with distilled water, dehydrated in an ethanol series followed by propylene oxide, infiltrated and finally embedded in Epon resin. Serial 50 nm sections were cut using a Diatome diamond knife and a Leica UC6 ultramicrotome, and picked up on Pioloform support films with 2 nm C on Synaptek slot grids. Sections were stained with uranyl acetate followed by Sato's lead (*Sato, 1968*) prior to imaging. An FEI Spirit TEM operated at 80kV was used to image the serial sections at 2.3 × 2.3 nm pixel resolution using Leginon (*Suloway et al., 2005*).

## L3v image volume registration

The L3v consisted of ≈ 300,000 4k × 4k image tiles, which were montaged and aligned using linear and nonlinear methods (*Saalfeld et al., 2012*) in TrakEM2 (*Cardona et al., 2012*). Filters for brightness and contrast correction were applied before montaging (Default min and max, normalized local contrast, enhance contrast). Images were first montaged in a section with two passes of linear montaging, first targeting only a translation transformation, and in the second pass targeting an affine transformation. This was followed by an elastic, non-linear montaging pass. For alignment between sections, parameter exploration was performed on a scaled down substack (scale factor 10) of 5 sections, targeting extraction of approximately 2000 features, 100 correspondences and an average displacement of 10 pixels. Linear alignment was applied to all sections using an affine transformation model. The "Test Block Matching Parameters' tool (http://imagej.net/Test_Block_Matching_Parameters) was used on five adjacent sections to find optimal parameters for the elastic registration pass. Elastic alignment was applied with local smoothness filter approximating an affine local transformation. The resulting aligned image stack was exported to an image tile pyramid with six scale levels for browsing and circuit reconstruction in CATMAID (*Saalfeld et al., 2009*; *Schneider-Mizell et al., 2016*). The L3v image stack is available at https://neurodata.io/.

## Neuron reconstruction

For the annotation of mdIV targets in the L1v, we manually reconstructed all neurons pre- and post-synaptic to the previously-described mdIV terminals in segment A1 (*Ohyama et al., 2015*). Circuit reconstruction in both datasets was performed in CATMAID following annotation and review procedures described previously (*Schneider-Mizell et al., 2016*). For 1004/1096 post-synaptic connections and 85/85 pre-synaptic connections, we were able to reconstruct an identifiable neuron. This included 173 neurites spanning a total of 30.2 mm in cable length, 13,824 synaptic inputs, and 18,624 synaptic outputs. For each cell type that exhibited more than 3 synapses of input from or output onto mdIV terminals on both left and right sides of the body, we fully reconstructed and comprehensively reviewed a left and right pair of neurons. No unpaired medial neurons were found. For segmentally repeated cell types that exhibited multiple segments of connection, we chose to review examples from the segment with the most synapses from mdIVs, typically segment A1. Reconstructions here were performed by CMSM (30.2%, 104,335/345,917 nodes), IA (28.0%, 96,999/345,917 nodes), Javier Valdes Aleman (8.7%, 29,942/345,917 nodes), Laura Herren (8.1%, 28,165/345,917 nodes), Waleed Osman (7.1%, 24,727/345,917 nodes), and 3% or less each from several other contributors. Comprehensive reviews of arbors and synapses in the L1v were performed by AC and CMSM.

For annotation of the new L3v, we specifically targeted mdIV axons in segment A3 using characteristic anatomical features, particularly entry nerves and the ventromedial location of pre-synaptic boutons. This segment was selected for its centrality in the EM volume and lack of section gaps. Interneurons were identified based on cell body location, neuropil entry point of the primary neurite and characteristic branching structures. To identify target cells from imagery, the principal branches of candidates were reconstructed until they could be conclusively identified from characteristic features. The reconstruction of 6 mdIV terminals and 12 specific LNs spanned 15.3 mm, 10035 synaptic

inputs, and 13499 outputs. Reconstructions were performed by IA (50.8%, 91,836/180,753 nodes), SG (21.9%, 39,654/180,753 nodes), CMSM (19.3%, 34,927/180,753 nodes), AC (6.4%, 11,611/180,753 nodes), and Waleed Osman (1.5%, 2,725/180,753 nodes). Comprehensive reviews of arbors and synapses in the L3v were performed by SG, AC and CMSM.

## Analysis

Neurons were exported from CATMAID (*Saalfeld et al., 2009*; *Schneider-Mizell et al., 2016*) through custom python scripts and imported into python or MATLAB (The Mathworks, Inc.) environments for analysis. Analysis was performed with custom MATLAB scripts with statistics performed using SciPy and R. Morphology and connectivity data were exported from CATMAID and imported into Matlab as a custom neuron data structure to ease analysis. Neuron data structures contained the spatial and topological information for every skeleton node in reconstructions, as well as their polyadic synapses, and annotations such as the location of twig roots and cell bodies. The data structures permitted network-based analysis and visualization using custom scripts and the Brain Connectivity Toolbox (*Rubinov and Sporns, 2010*). Analysis scripts and files describing neuronal morphology, synapse locations and connectivity can be found at https://github.com/ceesem/Larva_development_structure_2017 (*Schneider-Mizell, 2017*). A copy is archived at https://github.com/elifesciences-publications/Larva_development_structure_2017.

Neurons were split into axonal and dendritic compartments to maximize spatial segregation along the arbor between synaptic inputs and outputs using previously describes algorithms (*Schneider-Mizell et al., 2016*). The synaptic segregation index ($S$) was defined as before (*Schneider-Mizell et al., 2016*):

$$S = -\frac{1}{S_0(N_{ax}+N_{den})}\sum_{i=ax,den}N_i(log(p_i)+log(1-p_i))$$

where $N_i$ is the number of synaptic contacts in compartment $i$ (either axon or dendrite), $p_i$ is the fraction of synaptic contacts that are inputs, and $S_0 = -(log(p)+log(1-p))$ for $p$ being the fraction of all synaptic contacts that are inputs (*Schneider-Mizell et al., 2016*). $S_0$ is the maximum possible value of $S$ for a fully unsegregated neuron with the same numbers of synaptic inputs and outputs.

For the receptive field orientation analysis, we defined six unit vectors in a 2D plane $\hat{u}_j = \cos(\theta_j)\hat{x} + \sin(\theta_j)\hat{y}$, with the angle $\theta_j$ corresponding to the approximate center of each of the mdIV terminals ($\theta_j = j\pi/3$, with v'ada R corresponding to $j = 0$ and the mdIVs ordered counterclockwise). The mean orientation of interneuron $i$, $\vec{r}_i$, was computed as

$$\vec{r}_i = \frac{1}{\sum_{j=0}^{5}A_{ij}}\sum_{j=0}^{5}A_{ij}\hat{u}_j$$

where $A_{ij}$ is the number of synapses from mdIV neuron $j$ to LN $i$ and the sums are over all six mdIVs.

For the filling fraction analysis, we computed potential synapses for a connection from an mdIV terminal onto an LN by computationally removing all terminal branches (Strahler order 1) from the LN dendrites and measuring the number of pre-synaptic sites that were within a distance ($d = 2\mu m$ unless specified) of the arbor. This approximates a distance that could feasibly be spanned by typical twig growth without overestimating a neuron's spatial extent.

For the random twig omission errors for a given mdIV→LN connection, we assumed that each twig could be omitted with an independent probability $p$. We generated 5000 random instances for each LN and value of $p$. The synaptic counts were computed by considering only synaptic connections on remaining, non-removed twigs.

## Acknowledgements

We thank Stephan Saalfeld for assistance registering the L3v, Maarten Zwart, Matthias Landgraf, Chris Q Doe, and Marta Zlatic for helpful comments, and James Truman for generously sharing light microscopy data to help identify neurons. This work was funded by the Howard Hughes Medical Institute.

## Additional information

### Funding

| Funder | Author |
| --- | --- |
| Howard Hughes Medical Institute | Stephan Gerhard<br>Ingrid Andrade<br>Richard D Fetter<br>Albert Cardona<br>Casey M Schneider-Mizell |

The funders had no role in study design, data collection and interpretation, or the decision to submit the work for publication.

### Author contributions

Stephan Gerhard, Conceptualization, Data curation, Investigation, Visualization, Writing—review and editing; Ingrid Andrade, Investigation; Richard D Fetter, Resources, Investigation, Writing—review and editing; Albert Cardona, Conceptualization, Supervision, Investigation, Methodology, Writing—review and editing; Casey M Schneider-Mizell, Conceptualization, Data curation, Formal analysis, Investigation, Visualization, Writing—original draft

### Author ORCIDs

Stephan Gerhard (iD) http://orcid.org/0000-0003-4454-6171

Albert Cardona (iD) https://orcid.org/0000-0003-4941-6536

Casey M Schneider-Mizell (iD) http://orcid.org/0000-0001-9477-3853

### Decision letter and Author response

Decision letter https://doi.org/10.7554/eLife.29089.018

Author response https://doi.org/10.7554/eLife.29089.019

## Additional files

### Supplementary files

• Supplementary file 1. Atlas of all cell types synaptically connected to mdIVs. For each cell type, we show a dorsal view (with CNS boundary, anterior up), a sagittal view (anterior to right), a cross-sectional view (grey line indicates neuropile boundary), and a table of number and fraction (in parentheses) of synapses from mdIV neurons onto the neuron shown. Due to varying anteroposterior extents of neurons, sagittal views are not to scale.

DOI: https://doi.org/10.7554/eLife.29089.015

• Transparent reporting form

DOI: https://doi.org/10.7554/eLife.29089.016

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
