## [Decision Letter]

Thank you for submitting your article "Conserved neural circuit structure across *Drosophila* larva development revealed by comparative connectomics" for consideration by *eLife*. Your article has been favorably evaluated by Eve Marder (Senior Editor) and three reviewers, one of whom, Ronald L Calabrese (Reviewer #2), is a member of our Board of Reviewing Editors. The following individual involved in review of your submission has agreed to reveal their identity: Hermann Cuntz (Reviewer #1).

The reviewers have discussed the reviews with one another and the Reviewing Editor has drafted this decision to help you prepare a revised submission.

Summary:

This is a strong advance on an important paper in *eLife* in 2016, which reported new methods to apply quantitative arbor and network context to iteratively proofread and reconstruct circuits and create anatomically enriched wiring diagrams in *Drosophila*. They used these methods to measure the morphological underpinnings of connectivity in new and existing reconstructions of *Drosophila* sensorimotor (larva 1st instar L1v) circuits in the brain and VNC. They extend this work here by similarly reconstructing nociceptive networks in segment A3 of the VNC in new [encompassing segment A3 of the third instar larva (L3v)] and existing serial section electron microscopy volumes. They report a remarkable conservation of connectivity (topographically-arranged circuit structure) across instars despite considerable neuronal growth and increases in total number of synapses. Five-fold increases in size of interneurons were associated with compensatory structural changes that maintained cell-type-specific synaptic input as a fraction of total inputs. They argue that their new data on A3 VNC in L3V validates the use of the L1V total nervous system volume to determine connectivity across larval stages, and that patterns of structural growth in larval development act to conserve the computational function of a circuit, for example determining the location of a nociceptive stimulus. The work is carefully done with appropriate quantification and statistics. The connectome established and the atlas of neurons will be very useful for fly workers who wish to pursue behavioral or developmental studies.

Essential revisions:

1) The study is written in a very compact manner and would gain from substantial rearranging of the story and adding descriptive text. The story constantly switches between mdIVs and LNs, and it is very difficult to understand why. All sections of the text are too short.

2) The Materials and methods section on the data analysis is completely missing.

3) The figures need some rethinking and corrections. The terms/labels in the figures are often not described (not even used) in the main text, which makes them difficult to understand. Some of the supplemental figures could also be incorporated into the main article (e.g. Figure 2—Figure supplement 3 and Figure 2—figure supplement 4). Many figure panels mentioned in the text are entirely missing in the figures.

---

## [Author Response]

Essential revisions:1) The study is written in a very compact manner and would gain from substantial rearranging of the story and adding descriptive text. The story constantly switches between mdIVs and LNs, and it is very difficult to understand why. All sections of the text are too short.

We were pleased to have the room to expand the manuscript. More descriptive text has been added to nearly every paragraph discussing results. We believe the revised manuscript motivates its structure considerably more and will be clearer to the reader.

2) The Materials and methods section on the data analysis is completely missing.

We have greatly expanded the data analysis section of Materials and methods to detail our representation of neuronal reconstructions, filling fraction analysis, receptive field analysis, and error simulation. We have also posted analysis scripts on Github.

3) The figures need some rethinking and corrections. The terms/labels in the figures are often not described (not even used) in the main text, which makes them difficult to understand. Some of the supplemental figures could also be incorporated into the main article (e.g. Figure 2—figure supplement 3 and Figure 2—figure supplement 4). Many figure panels mentioned in the text are entirely missing in the figures.

We have addressed these concerns by expanding the number of figures, making terminology more consistent and clear, and fixing typos in the text figure references. Throughout all figures, we increased text size and spacing of panels and labels. Major changes included:

Previous Figure 2 was split into two figures – Figure 2 and Figure 3 –, allowing clearly presentation of results and more consistent;New Figure 3 includes key results on the topographical structure of sensory input from the previous Figure 2—figure supplement 3;Previous Figure 2—figure supplement 4 is now Figure 4;Previous Figure 3 and Figure 4 are now revised Figure 5 and Figure 6, respectively.

All figure panels referenced in the text that were previously missing in figures were reference errors and intended to point to panels in the original set of figures. We have carefully checked figure references in the revised manuscript.

As an additional point, on re-evaluation of the data in response to reviewer comments, we also noticed that two cell types (A09a and A09c) in the L3v had axons that were truncated by the volume boundary. As a result, we removed them from analysis that required a complete count of the cable length or synapse count (Figure 2 as well as parts of Figure 5). While some of the axon-related results still held, we de-emphasized them in the text and moved figure panels regarding axon size to Figure 2—figure supplement 2. Because the primary subject of this work is LN dendritic development, we believe this has little impact on the overall story.